# Chondroprotective Effects of Chondrogenic Differentiated Adipose-Derived Mesenchymal Stem Cells Sheet on Degenerated Articular Cartilage in an Experimental Rabbit Model

**DOI:** 10.3390/bioengineering10050574

**Published:** 2023-05-10

**Authors:** Atsushi Taninaka, Tamon Kabata, Katsuhiro Hayashi, Yoshitomo Kajino, Daisuke Inoue, Takaaki Ohmori, Ken Ueoka, Yuki Yamamuro, Tomoyuki Kataoka, Yoshitomo Saiki, Yu Yanagi, Musashi Ima, Takahiro Iyobe, Hiroyuki Tsuchiya

**Affiliations:** Department of Orthopaedic Surgery, Graduate School of Medical Sciences, Kanazawa University, Ishikawa 920-8641, Japan; atsushi880628@yahoo.co.jp (A.T.);

**Keywords:** adipose-derived mesenchymal stem cells, cell sheet, chondrocyte, osteoarthritis, regenerative therapy

## Abstract

Adipose-derived stem cells (ADSCs) have been studied for many years as a therapeutic option for osteoarthritis (OA); however, their efficacy remains insufficient. Since platelet-rich plasma (PRP) induces chondrogenic differentiation in ADSCs and the formation of a sheet structure by ascorbic acid can increase the number of viable cells, we hypothesized that the injection of chondrogenic cell sheets combined with the effects of PRP and ascorbic acid may hinder the progression of OA. The effects of induction of differentiation by PRP and formation of sheet structure by ascorbic acid on changes in chondrocyte markers (collagen II, aggrecan, Sox9) in ADSCs were evaluated. Changes in mucopolysaccharide and VEGF-A secretion from cells injected intra-articularly in a rabbit OA model were also evaluated. ADSCs treated by PRP strongly chondrocyte markers, including type II collagen, Sox9, and aggrecan, and their gene expression was maintained even after sheet-like structure formation induced by ascorbic acid. In this rabbit OA model study, the inhibition of OA progression by intra-articular injection was improved by inducing chondrocyte differentiation with PRP and sheet structure formation with ascorbic acid in ADSCs.

## 1. Introduction

Osteoarthritis (OA) is a chronic degenerative joint disease that affects millions of people worldwide, and causes pain, stiffness, and disability [1,2]. OA is characterized by progressive loss of articular cartilage, subchondral bone remodeling, and synovial inflammation, leading to joint destruction and functional impairment [3,4,5]. In the past, the optimal treatment for patients with OA was focused on joint pain relief and joint function improvement [6,7,8].

Most patients with OA are treated conservatively with nonsteroidal anti-inflammatory drugs (NSAIDs) and other analgesics, intra-articular injections of hyaluronic acid and corticosteroids, and physical therapies; however, they have limited effectiveness [7,8]. The use of analgesics is also associated with side effects, such as gastrointestinal bleeding, renal toxicity, and cardiovascular events [9].

Although surgical interventions, such as joint arthroplasty and osteotomies, have been performed in many patients to relieve joint pain and improve joint function, these are invasive procedures that carry the risks of complications, infection, and implant failure [10,11]. Moreover, joint arthroplasty may not be suitable for younger patients or those with comorbidities that interfere with surgery.

Regenerative medical approaches aiming to restore damaged tissues and promote tissue regeneration have emerged as promising alternatives for the treatment of OA. The goal of regenerative medicine is to harness the natural healing mechanisms of the body and enhance them using biological materials, cells, or growth factors. In recent years, various regenerative therapies, including stem cell, platelet-rich plasma (PRP), and gene therapies, have been investigated for their potential to regenerate articular cartilage and alleviate OA symptoms.

Inducing the differentiation of stem cells into chondrocytes and their use in regenerating cartilage tissue are the major goals of regenerative medicine. Bone marrow stromal cells, embryonic stem cells, and induced pluripotent stem cells have been studied in regenerative medicine and are considered promising stem cells. Additionally, adipose-derived stem cells (ADSCs), a type of mesenchymal stem cell, have recently attracted considerable attention [12,13,14,15,16]. Among the various sources of mesenchymal stem cells, ADSCs are an abundant source of multipotent adult stem cells that can be easily isolated from subcutaneous adipose tissue and cultured in large quantities using a minimally invasive procedure [17,18,19,20].

There are several methods for using ADSCs to repair or reconstruct injured tissues and organs. The classic application of ADSCs is through direct injection into the repair site [21]. The use of scaffolds composed of natural biodegradable matrices is an attractive strategy for overcoming the major limitation of stem cell therapy, which is the drawback of cell engraftment [21]. The use of ADSCs sheets in tissue regeneration has been widely reported.

Ascorbate-2-phosphate has been reported to increase the secretion of collagen proteins by mesenchymal cells, but has no effect on cells [22]. However, collagen is considered to be a good natural carrier for chondrogenic cell sheets. Furthermore, it has been reported that administering ADSC sheets, which have been formed by treating ADSCs with ascorbic acid to create a sheet structure, may inhibit the progression of OA, which may be due to the large number of ADSCs implanted through sheet structure formation [23].

There have been scattered reports that direct intra-articular injection of PRP can relieve pain because PRP contains many growth factors and other cytokines; however, PRP also has the potential to induce chondrocyte-specific genetic changes in ADSCs [24,25,26]. The possibility of inducing chondrogenic differentiation using adipose-derived stem cells, which can be easily harvested and cultured, and PRP, which can be relatively easily prepared from peripheral blood, has been reported, and more therapeutic effects may be expected by forming a sheet structure with the effect of ascorbic acid on differentiated induced chondrogenic cells.

Therefore, this study aimed to improve the regeneration of degenerated articular cartilage and inhibit OA progression in a rabbit anterior cruciate ligament transection (ACLT)-induced OA model, using chondrogenic cell sheets that combine the PRP differentiation of ADSCs into chondrocytes and the effect of ascorbic acid sheet formation on ADSCs.

## 2. Materials and Methods

### 2.1. Experimental Animals

Sixty skeletally mature female Japanese white rabbits, weighing 2.5–3.0 kg, were used for this experiment. The Institute for Experimental Animals, Kanazawa University Advanced Science Research Center approved all experimental protocols using an animal model. All surgical procedures were performed in accordance with the Guide for the Care and Use of Laboratory Animals published by the United States National Institutes of Health (Bethesda, MD, USA; NIH publication no. 86–23, revised in 1985).

### 2.2. Establishment of a Rabbit Osteoarthritis Model

OA was induced as described in a previous study [27]. Medetomidine hydrochloride and midazolam were intramuscularly injected to anesthetize the rabbits, and their knees were shaved and sterilized with iodine. The right knee joint was exposed through a medial parapatellar incision, the patella was dislocated laterally to access the anterior cruciate ligament (ACL), and the knee was placed in the full flexion position. The ACL was optimally visualized and transected with microscissors, and a positive anterior drawing test was performed to confirm complete ligament transection (Figure 1). The joints were irrigated and sutured routinely. The same procedure was performed on the left knee. Postoperatively, the rabbits were allowed free activity in their cages without immobilization. Traumatic degeneration was induced as previously described for the ACLT model [28], which is characterized by OA-like damage.

### 2.3. Experimental Grouping

The rabbits used in the experiment were divided into the following four groups according to the administered cell groups: (1) ADSC group (control), (2) ADSC sheet group with sheet structure induced by ascorbic acid, (3) PRP-induced chondrogenesis group with ADSCs (PRP-ADSCs) group, and (4) ADSCs induced to chondrogenic differentiation by PRP were then sheet-structured using the PRP-ADSC sheet group.

### 2.4. Isolation of Adipose-Derived Stem Cells

ADSCs were isolated as previously described [29]. Approximately 1–2 g of subcutaneous adipose tissue from the posterior neck region of the rabbit was harvested, and the visible blood vessels and fascia tissue were removed and washed in phosphate-buffered saline (PBS; Wako Pure Chemical Industries, Osaka, Japan). After cutting the adipose tissue into small strips, the adipose tissue was digested in PBS dissolving collagenase (Wako Pure Chemicals) at a concentration of 0.12% for 45 min at 37 °C using a water bath. During the digestion, the mixture was shaken for 15 min. Immediately after digestion, Dulbecco’s modified Eagle’s medium (DMEM; Wako Pure Chemical Industries) was added to neutralize collagenase activity. The digested samples were filtered and centrifuged at 1300 revolutions per minute (rpm) for 6 min at 25 °C to remove the supernatant, and the cells were cultured in high glucose DMEM, supplemented with 10% fetal bovine serum (FBS; Nichirei Biosciences, Tokyo, Japan) and 1% penicillin-streptomycin solution (P/S; Wako Pure Chemical Industries) in a 5% CO_2_ incubator at 37 °C. After 24 h, the cells were washed with PBS to remove debris, and then a fresh medium was added. After 3–4 days, spindle-shaped cell attachment was observed. Subsequently, after the seeded cells proliferated and reached 80–90% confluence, the cells were trypsinized and isolated as ADSCs. The third passage of ADSCs was used for subsequent experiments. The presence of stemness in ADSCs extracted by this method has been confirmed in previous reports [30].

### 2.5. Fabrication of Adipose-Derived Stem Cell Sheets

ADSC sheets were fabricated following a previously reported protocol [31]. To generate cell sheets, ADSCs were seeded at a density of 1 × 10^6^ cells/dish in 100 mm tissue culture dishes and cultured for 7 days in a medium consisting of DMEM, 10% FBS, 1% P/S, and 50 mM ascorbate-2-phosphate (Vitamin C). The culture medium was renewed every 2–3 days over the course of 1 week. The fabrication medium, which contained ascorbate-2-phosphate, was used to promote the secretion of collagen by ADSCs and to facilitate the formation of ADSC sheets (Figure 2).

### 2.6. Chondrogenic Differentiation of Adipose-Derived Stem Cells Induced by Platelet-Rich Plasma

#### 2.6.1. Preparation and Activation of Platelet-Rich Plasma

Blood samples were obtained from Japanese white rabbits weighing 2.5–3.0 kg. PRP was enriched using a two-step centrifugation process [32]. Eighteen mL of whole blood was drawn from the central auricular artery of each rabbit into two sterile tubes, each containing 1 mL of acid citrate dextrose-A solution as an anticoagulant. The tubes were centrifuged at 250× *g* for 7 min (himac CF16RX, Hitachi Koki Co., Ltd., Tokyo, Japan) at 25 °C, and the blood was separated into three phases: platelet-poor plasma (top), platelet-rich plasma (middle), and erythrocytes (bottom). The upper plasma layer and middle buffy coat were collected in separate sterile tubes, taking care not to mix the red blood cells to minimize red blood cell interference. The collected upper and middle layers were centrifuged at 350× *g* for 10 min. Approximately 2 mL of sediment and plasma were collected from the bottom to obtain PRP. All products were activated with a one-tenth volume of 10% CaCl_2_ solution and incubated overnight at 37 °C; then, they were centrifuged at 1000× *g* for 10 min to collect the supernatant to obtain the activated PRP.

#### 2.6.2. Chondrogenic Differentiation

ADSCs at the third passage were used to induce chondrogenic differentiation by culturing them in a medium containing 10% PRP by volume, as described previously [33]. The culture medium was replaced every 3–4 days for a week duration.

### 2.7. Fabrication of Platelet-Rich Plasma (PRP)-Adipose-Derived Stem Cell Sheets

PRP-ADSC sheets were generated using the same method as regular ADSCs. The sheet structure was obtained by culturing ADSCs that had been induced into chondrogenic differentiation using PRP with ascorbic acid for 7 days (Figure 3).

### 2.8. In Vitro Study

#### 2.8.1. Quantitative Reverse Transcription-Polymerase Chain Reaction Analysis

The mRNA levels of representative chondrocyte markers, type II collagen, aggrecan, and Sox9, were examined in the ADSCs, ADSC sheet, PRP-ADSCs, and PRP-ADSC sheet groups using quantitative reverse transcription-polymerase chain reaction (qRT-PCR). Ten samples from each group were homogenized and RNA was extracted using TRIzol reagent (Invitrogen) according to the manufacturer’s instructions. Briefly, each sample was placed in a sterile tube containing TRIzol reagent and incubated at 25 °C for 5 min. Next, 200 µL of pre-chilled chloroform per ml of TRIzol was added to the tube, the solution was stirred, and incubated at 25 °C for 10 min. Samples were centrifuged at 4 °C, and the upper aqueous phase (500 μL) was transferred to an RNase-free centrifuge tube and incubated with an equal volume (500 μL) of isopropanol before centrifugation at 25 °C. The precipitated RNA pellet was washed with 500 μL of 80% (*v/v*) ethanol, after which the sample was centrifuged for 5 min and the supernatant was discarded.

For reverse transcription, 6 µg of mRNA was incubated with a RevertAid First-Strand cDNA Synthesis Kit (T100TM Thermal Cycler; Bio-Rad, Hercules, CA, USA) using a thermal cycler (T100TM Thermal Cycler; Bio-Rad, Inc. Thermo Fisher Scientific). Next, real-time RT-PCR was performed using SYBR Green PCR Master Mix (Applied Biosystems, Foster City, CA, USA) on a StepOne^TM^ system (Applied Biosystems), according to the manufacturer’s instructions. The amplification parameters were an initial 95 °C incubation step for 15 min, followed by 20 amplification cycles of 94 °C for 15 s, 60 °C for 30 s, and 72 °C for 30 s. The reactions ended with a 72 °C-extension step for 7 min, followed by storage at 4 °C overnight. The expression level of each target gene was calculated relative to that of glyceraldehyde 3-phosphate dehydrogenase (GAPDH) in each sample. The relative expression levels were calculated using the 2^−ΔΔCT^ method.

The following primers (Hokkaido System Science Co., Ltd., Hokkaido, Japan) were used for qRT-PCR analysis:

GAPDH (Rabbit)-F: 5′-CTTCGGCATTGTGGAGGGGC-3′;

GAPDH (Rabbit)-R: 5′-GGAGGCAGGGATGATGTTCT-3′;

Type Ⅱ collagen (Rabbit)-F: 5′-TCGGCCTCCCTGGTATTGACG-3′;

Type Ⅱ collagen (Rabbit)-R: 5′-GGAGGGCCCTGAGCACCATTGTT-3′;

Sox9 (Rabbit)-F: 5′-AAGGGCTACGACTGGACGCTGGTG-3′;

Sox9 (Rabbit)-R: 5′-AGGGCCGCTTCTCGCTCTCG-3′;

Aggrecan (Rabbit)-F: 5′-GTCTACAGAACAGCGCCATCATT-3′;

Aggrecan (Rabbit)-R: 5′-GCGAAGCAGTACACGTCATAGGT-3′.

#### 2.8.2. Enzyme-Linked Immunosorbent Assay

The changes in VEGF-A expression by the induction with PRP or by sheeting with ascorbic acid were also evaluated using enzyme-linked immunosorbent assay (ELISA). To determine the concentration of secreted VEGF-A, the culture supernatant of each group of samples was collected and then subjected to ELISA (LifeSpan BioSciences, Inc., Seattle, WA, USA). VEGF concentrations were also measured in fresh medium, and VEGF produced by ADSCs was calculated by subtracting the value in the culture supernatant from that in the fresh medium.

#### 2.8.3. Toluidine Blue Staining

The cartilage formation was evaluated by performing toluidine blue staining and assessing the secretion of aggrecan for each of the four different groups [34]. The culture period was adjusted prior to staining. Toluidine blue staining was performed twice at a 7-day interval. The cells were fixed with 400 μL of 75% paraformaldehyde (PFA) for 15 min at 25 °C before removing the fixing solution and washing the cells twice with 500 μL of PBS. The cells were then stained with toluidine blue (Solarbio G3661) for 30 min. Excess dye was removed and the cells were washed three times with PBS before imaging with an optical microscope (Olympus, CKX31). Proteoglycans, such as aggrecan secreted from chondrocytes into the cartilage matrix, were stained purple with toluidine blue; therefore, increased purple intensity indicates the presence of more secreted proteoglycans.

### 2.9. In Vivo Study

#### 2.9.1. Injection of Each Cell Group

Autologous ADSCs, ADSC sheets, PRP-ADSCs, and PRP-ADSC sheets were prepared from subcutaneous adipose tissue and blood in conjunction with experimental OA induction. Cells and cell sheets were washed three times with PBS to wash out the culture medium. The cells and cell sheets were detached from the culture dishes using trypsin or a cell scraper. Four weeks after ACLT, cells or cell sheets of each group were injected intra-articularly into the rabbit’s right knee using an 18-gauge needle. The same cell group was injected into the left knee. At 6-, 8-, 10-, and 12 weeks after ACLT, the rabbits were sacrificed (3 rabbits/6 knees at each period), and the femoral condyles of both knees were harvested (Figure 4).

#### 2.9.2. Macroscopic Analysis

To evaluate cartilage damage, India ink (American MasterTech, Lodi, CA, USA) was used to stain the femoral condyles and macroscopic images were captured using a SONY α6100 digital camera (SONY Corp., Tokyo, Japan). The visual observations were classified and scored based on previously reported criteria [35,36]. The medial and lateral femoral condyles were separately scored on a scale of 0 (intact joint surface) to 5 (erosion of 5 mm or more) according to the degree of fibrosis and erosion (Table 1). The scores for both condyles were combined to obtain a cumulative score for macroscopic OA. All evaluations were performed by two blinded researchers and the scores were averaged to calculate the overall score.

#### 2.9.3. Histologic Analysis 

After macroscopic examination, dissected distal femurs were fixed in a 4% paraformaldehyde solution and subsequently decalcified in a 4% ethylenediaminetetraacetic acid (EDTA) solution. The specimens were dehydrated in a gradient ethanol series, embedded in paraffin blocks, and sectioned into 5 μm slices. Safranin O staining was used to evaluate the general morphology and proteoglycan/collagen content of the cartilage. The tissue sections were visualized under a fluorescence microscope (Keyence, Osaka, Japan). Ten sections were created along the coronal plane passing through the center of the femoral condyle, and one section from each sample, including the most severely degenerated part, was used for each histological analysis. Two blinded researchers evaluated the severity of cartilage degeneration using the Osteoarthritis Research Society International (OARSI) Cartilage OA Histopathological Grading System [37].

#### 2.9.4. Immunohistochemical Analysis

Immunohistochemical staining was performed to assess the expression of catabolic enzymes involved in the breakdown of cartilage matrix proteins, as previously described [38]. Tissue sections were deparaffinized and treated with hyaluronidase, followed by incubation with mouse anti-human MMP-1 monoclonal antibody (1:100; Kyowa Pharma Chemical Co., Toyama, Japan), mouse anti-rabbit MMP-13 monoclonal antibody (1:20; Thermo Fisher Scientific Inc., Waltham, MA, USA), and mouse anti-human ADAMTS-4 monoclonal antibody (1:150; Thermo Fisher Scientific Inc. software (Waltham, MA, USA) at 25 °C. Antibodies were diluted using an antibody diluent (Agilent Technologies, Santa Clara, CA, USA). The sections were then incubated with labeled polymer-HRP anti-mouse IgG (Dako, Tokyo, Japan) for 30 min at 25 °C after overnight incubation with primary antibody at 4 °C. The signal was detected using 3,3′-diaminobenzidine tetrahydrochloride and the nuclei were counterstained with hematoxylin. Semi-quantitative analysis was performed on six microscopic fields (×100 magnification) representing the inner, central, and outer regions of the cartilage tissue. A semi-quantitative method was employed to assign values to the immunohistochemical results as the percentage of positive cells (MMPs and ADAMTS-4), with a maximum score of 100% for a comprehensive evaluation of protein expression [39]. Immunohistochemical results were assessed by two observers who were blinded to the identity of each sample.

#### 2.9.5. Labeling Using 10-Dioctadecyl-3,3,30,30-Tetramethylindocarbocyanine Perchlorate 

To observe the fate of the ADSCs after intra-articular injection, 10-dioctadecyl-3,3,30,30-tetramethylindocarbocyanine perchlorate (DiI) labeling was performed. Sixteen rabbits (32 knees) were used in this study. 

ADSCs, ADSC sheets, PRP-ADSCs, and PRP-ADSC sheets were labeled with the fluorescent lipophilic tracer DiI (Molecular Probes, Eugene, OR, USA) according to the manufacturer’s instructions to track the cells. Four weeks after ACLT, rabbits were divided into four groups and injected with labeled cells into both knees. At 6-, 8-, 10-, and 12 weeks after ACLT, the rabbits were sacrificed and the knee joints were dissected. Freshly frozen sagittal sections, including the central part of the knee joint, were prepared using Kawamoto’s method [40]. The stained slides were imaged with a SLIDEVIEW VS200 (EVIDENT Co., Ltd., Tokyo, Japan) research slide scanner.

### 2.10. Statistical Analysis

All data were analyzed using the Statistical Package for the Social Sciences (SPSS) software (version 25.0; SPSS, Inc., Armonk, NY, USA). Multiple groups were compared using Welch’s analysis of variance (ANOVA), followed by Games-Howell post-hoc tests. The chi-square test was used to evaluate the comparisons of proportions among the groups. Normally distributed data are expressed as mean ± standard error or mean ± standard deviation. Statistical significance was set at *p* < 0.05.

## 3. Results

### 3.1. Association of Platelet-Rich Plasma and Ascorbic Acid with Adipose-Derived Stem Cells, Chondrocyte Marker Expression

The expression levels of each gene in the four groups (*n* = 10) were analyzed using qRT-PCR, and similar results were observed for type II collagen, aggrecan, and Sox9 (Figure 5). Incubation in the PRP-containing medium resulted in a considerable increase in the expression of each gene. Although there was a significant difference in mRNA expression between the ADSC-sheet and PRP-ADSC sheet groups, there was no difference in expression between the PRP-ADSCs and PRP-ADSC sheet groups.

### 3.2. Association of Platelet-Rich Plasma with VEGF-A Secretion

Ten samples from each group were used to measure the concentration of VEGF-A using ELISA. Our findings showed a decrease in VEGF-A production when the cells were cultured in a medium containing PRP (Figure 6).

In the ADSCs group, the concentration of VEGF-A was 182.17 ± 19.63 ng/mL, whereas, in the ADSC sheet group, it was 167.44 ± 54.34 ng/mL; PRP-ADSCs 135.03 ± 63.01 ng/mL; and PRP-ADSC sheet, 111.80 ± 38.34 ng/mL. The PRP-ADSC and PRP-ADSC sheet groups exhibited a significant decrease in VEGF-A concentration compared with the ADSCs group. Furthermore, the PRP-ADSC sheet group displayed a notable decrease compared with the ADSC sheet group. However, no significant differences were observed between PRP-ADSCs and PRP-ADSC sheets (*p* = 0.33).

We detected a significant decrease in the concentration of VEGF-A from ADSCs after culturing in a PRP-containing medium, which was not impaired by the addition of ascorbic acid. Therefore, our results suggest that the inhibitory effect of VEGF-A on cartilage regeneration may be suppressed in PRP-treated ADSC sheets.

### 3.3. Induction of Differentiation in Platelet-Rich Plasma-Containing Medium on Secretion of Glycans from Adipose-Derived Stem Cells

Toluidine blue staining revealed stronger purple staining in the two cell groups induced to differentiate with PRP (Figure 7). This result indicates that culturing ADSCs in a medium containing PRP leads to increased secretion of glycans, such as aggrecan, compared to culturing them in a normal medium. Additionally, the ADSC sheet group showed slightly stronger purple staining than the ADSCs sheet group. Glycan secretion was also observed in the presence of ascorbic acid. In the comparison between the PRP-ADSCs and PRP-ADSC sheet groups, although there was no significant difference in purple intensity, a slight enhancement was observed, which indicates an increase in the secretion of glycans into the substrate and a possible differentiation into chondrocytes that secrete mucopolysaccharides.

### 3.4. Macroscopic Osteoarthritis Evaluation

Figure 8 shows the macroscopic OA scores, which were significantly lower in the PRP-ADSC sheet group than in the other groups at 10- and 12 weeks after ACLT. Lower scores indicate less damage to the articular cartilage surface. The PRP-ADSC sheet groups also showed a decrease in score from as early as 6 weeks, suggesting that PRP-ADSC sheets may have an inhibitory effect on the progression of OA from an early stage.

### 3.5. Histological Osteoarthritis Evaluation 

Histological findings showed severe articular cartilage loss progression and decreased safranin-O staining intensity in the ADSCs group, whereas the PRP-ADSC sheet group showed less cartilage loss and less decrease in safranin-O staining intensity. The OARSI score at 12 weeks was significantly lower in the PRP-ADSC sheet group compared to the other groups (Figure 9). A decreased OARSI score was also observed as early as 6 weeks, suggesting an early protective role of the PRP-ADSC sheet on the articular cartilage structure.

### 3.6. Evaluation of the Effects of Platelet-Rich Plasma and Ascorbic Acid on the Expression of MMP-1, MMP-13, and ADAMTS-4

Degradation of the cartilage matrix is a crucial factor in the development and progression of OA. Therefore, we aimed to investigate the impact of PRP-ADSC sheet treatment on the catabolic and inflammatory molecules involved in OA progression, particularly the enzymes MMP-1, MMP-13, and ADAMTS-4, which degrade the extracellular matrix. To evaluate the expression of these enzymes, immunohistochemical staining was performed on samples obtained 8 weeks after ACLT.

The percentage of MMP-1-positive cells was significantly decreased in both the ADSC sheet and PRP-ADSC sheet groups, with a significant difference compared to the ADSC and PRP-ADSC groups. However, no significant differences were observed between the ADSC and PRP-ADSC sheets (*p* > 0.05). These findings suggest that ascorbic acid may play a role in inhibiting MMP-1 expression; whereas PRP did not appear to have a considerable effect.

Similar results were observed for MMP-13 and ADAMTS-4, with both ascorbic acid and PRP contributing to the downregulation of their expression. A significant downregulation was observed in the PRP-ADSC sheet group (Figure 10).

### 3.7. Periarticular Distribution of Cells in the PRP-ADSC Sheet Groups

Figure 11 shows the DiI-positive region, where the surviving cells were distributed in the periarticular synovial lining after intra-articular injection in each group. Although ADSCs showed positive areas at 6 weeks, no positive areas were identified after 8 weeks. The ADSC sheet group showed larger positive areas than the ADSCs group, and the PRP-ADSC sheet group had the longest duration of positive areas remaining. However, no positive areas suggesting cell engraftment to the articular surface were observed in either group.

## 4. Discussion

Stem cell therapy is a promising treatment option that lies between traditional conservative therapy and surgical treatment of OA. This option is becoming a new hope in the field of regenerative medicine for patients who cannot undergo surgical treatment or do not respond to traditional conservative therapy. However, stem cell therapy has not yet achieved sufficient results to meet expectations, and further research is required.

In this study, we evaluated the effectiveness of intra-articular injection of chondrogenically induced ADSC sheets in the treatment of knee OA using an experimental animal model. The effectiveness of ADSC administration for OA and the induction of ADSCs into chondrocytes using a culture medium containing PRP has been previously reported [6,39,41,42,43]. Furthermore, the addition of ascorbic acid to ADSCs can form a sheet structure, and a large number of ADSCs survive after the administration of this sheet [23]. This may be due to the sheet structure allowing cells to be transplanted as a continuous cell mass [44]. By combining the chondrogenic induction ability of PRP for ADSCs and the formation of sheet structures by ascorbic acid, we created chondrogenic-induced ADSC sheets. We anticipated the therapeutic effects of transplanting a large number of cells expressing chondrocyte genes. In addition, this sheet structure, which contains collagen and other components, shows stable adhesion to the surface of the cartilage matrix and is expected to adhere to damaged cartilage sites, which is a challenge in stem cell therapy for OA [21].

PRP can induce the differentiation of ADSCs into chondrocytes due to the presence of multiple types of growth factors [45]. Similar to previous reports, PRP-induced differentiation led to increased expression of characteristic chondrocyte genes such as type II collagen, aggrecan, and Sox-9. Furthermore, the expression was not diminished by the formation of sheet-like structures using ascorbic acid. This suggests that PRP changed the gene expression of ADSCs towards the chondrogenic lineage and that sheet structures could still be formed even with the addition of ascorbic acid, indicating that this sheet structure, which allows for the transplantation of continuous cells and prevents washout from the injection site, could improve the viability of transplanted cells. Therefore, it is expected that the administration of this chondrogenic cell sheet, which enables the administration of a large number of ADSCs with chondrogenic potential and improved engraftment capability, will lead to the repair of damaged articular cartilage [21].

Regeneration of damaged joint cartilage is inhibited by specific growth factors, such as VEGF-A, secreted from ADSCs. VEGF enhances catabolic pathways in chondrocytes and induces the expression of MMPs in immortalized chondrocytes [46]. Therefore, the excessive expression of VEGF is associated with the progression of OA. Lee et al. actually improved cartilage regeneration by neutralizing VEGF with a monoclonal antibody in ADSCs [47]. In previous studies, ADSCs have been shown to highly express VEGF-A and secrete it into the culture supernatant [47]. This VEGF-A-containing supernatant inhibits chondrocyte proliferation, decreases the mRNA levels of type 2 collagen and Sox9, decreases proteoglycan synthesis, and causes apoptosis. Therefore, ADSCs may inhibit cartilage regeneration, but the expression of VEGF-A was decreased by culturing the ADSCs in a medium containing PRP. In this study, we investigated the possibility of inhibiting VEGF expression by promoting the differentiation of ADSCs into chondrogenic cells using PRP and sheet formation using ascorbic acid. PRP treatment downregulated the secretion of VEGF in the culture supernatant of ADSCs, and although there was no considerable difference, treatment with ascorbic acid tended to downregulate VEGF secretion as well. PRP-treated ADSCs improve the repair of damaged articular cartilage to a greater extent than normal ADSCs [33]. The present results suggest that PRP-ADSC sheets may repair damaged articular cartilage even more effectively than PRP-ADSCs.

Chondrocytes secrete mucopolysaccharides such as aggrecan. Because aggrecan is stained with toluidine blue, the enhancement of purple staining intensity indicates the possibility of differentiation into chondrocytes [48]. In the present study, the secretion of aggrecan was increased by the treatment of ADSCs with PRP, and a similar or stronger enhancement was observed after ascorbic acid sheeting treatment, suggesting that PRP and ascorbic acid treatments enabled the formation of sheet structures containing chondrocytes.

The results of this study showed that OA progression was considerably less severe in the PRP-ADSC sheet group, both macroscopically and histologically. In contrast, no new cartilage formation was observed in either group. Although OA progressed over time in all groups, articular cartilage thickness was preserved the most in the PRP-ADSC sheet group. In this study, we used a rabbit model OA model induced by ACL transection because its pathology is similar to that of human OA [27,29,36]. Moreover, rabbits were selected for two reasons. First, mice and rats are too small to inject the created cell sheet, and it is difficult to create an OA knee in larger animals, such as pigs. Second, it was difficult to secure the required number of experimental animals. This experiment required relatively easy experimentation and collection of many specimens, making rabbits the optimal choice.

One possible mechanism underlying the effects of ADSCs in the treatment of damaged articular cartilage is that ADSCs injected into the joint may affect the periarticular microenvironment via trophic mechanisms by distributing growth factors and cytokines with chondroprotective effects [29]. The results obtained in this study showed that the expression of MMP-1, MMP-13, and ADAMTS-4 was the lowest in the PRP-ADSC sheet group among the four groups, and MMP-1 expression was decreased in the group treated with ascorbic acid. Evaluation of MMP-13 and ADAMTS-4 expression levels showed a decreased expression in both PRP-and ascorbic acid-treated groups, although no differences were observed in their respective effects. MMPs constitute a group of zinc-dependent proteases that degrade components of the extracellular matrix, including collagen, elastin, gelatin, and casein [49]. ADAMTS proteases are multidomain extracellular protease enzymes that process procollagen and cleave proteoglycans such as aggrecan, versican, brevican, and neurocan.

Both the MMP and ADAMTS families play important roles in the progression of OA [50]. MMP-1 and MMP13 are the major enzymes that degrade type II collagen, and ADAMTS-4 is the major enzyme that degrades aggrecan [51]. Both type II collagen and aggrecan are major components of the cartilage matrix. Our results suggest that PRP-ADSC sheets secrete more fluid factors with chondroprotective properties and inhibit the progression of articular cartilage degeneration.

The injection of undifferentiated ADSCs is a chondroprotective mechanism through the secretion of trophic factors with anti-inflammatory and other effects [6]. Treatment with ascorbic acid may influence the survival of ADSCs administered by the formation of a strong sheet structure, whereas PRP modifies the properties of ADSCs owing to the numerous growth factors and cytokines contained in the plasma.

In the present study, DiI-labeling of ADSCs showed the longest survival in the PRP-ADSC sheet group compared with the other groups. Consistent with a previous study, no adhesion to damaged articular cartilage was observed in either group; however, adhesion to the subintimal layer of the periarticular synovial membrane was observed in cells induced by PRP, which promoted cell differentiation and proliferation [23,29]. Surviving ADSCs may remain in the synovium and secrete fluid factors that have chondroprotective effects, such as the proliferation of chondrocytes and protection of the cartilage matrix, for a longer duration.

This study has some limitations. The animal species in the OA model used in our experiments were quadruped rabbits, whereas humans are bipeds; therefore, they may not reproduce the dynamics of the human knee, which may affect the progression of OA. Thus, further investigations are warranted in bipedal animals. Furthermore, there have been reports of individual variability in the progression of osteoarthritis (OA) four weeks after ACLT [52]. This may also have an impact on the evaluation of OA progression after cell injection in each group. Another limitation is that Toluidine blue staining was not quantitatively evaluated. It would be desirable to perform quantitative and objective evaluations in future studies. In addition, the number of cells was only standardized at the time of seeding in the third passage. However, PRP is reported to have cell proliferative and differentiation potential, and the number of cells injected may influence the assessment of cartilage damage [53]. Future studies are required to standardize the number of cells used at the time of administration to evaluate the effects of differentiation more accurately.

## 5. Conclusions

Chondrogenic cell sheets, in which ADSCs were induced to differentiate into chondrocytes by PRP and formed a sheet structure using ascorbic acid, may survive in the synovium for a long time, secreting liquid factors with protective effects on chondrocytes and the cartilage matrix, which may inhibit the progression of cartilage degeneration.

## Figures and Tables

**Figure 1 bioengineering-10-00574-f001:**
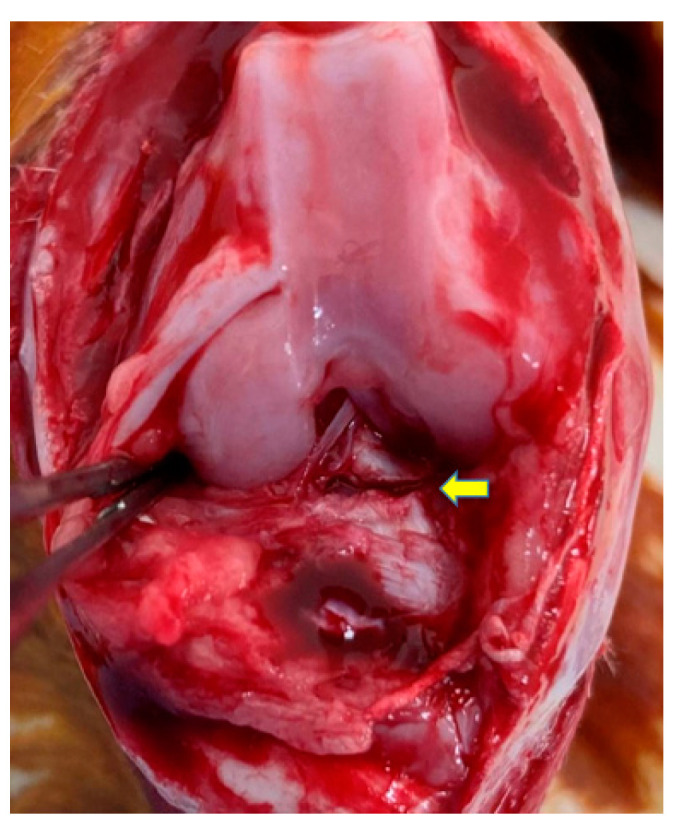
An anterior cruciate ligament (ACL) transection was performed to induce osteoarthritis (OA) change in the rabbit knee joint. The ACL was cut at the point indicated by the yellow arrow.

**Figure 2 bioengineering-10-00574-f002:**
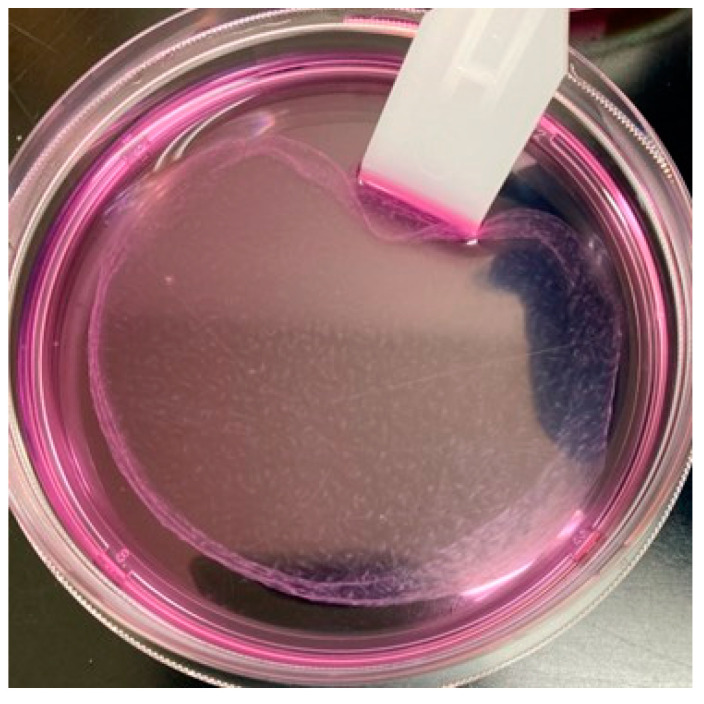
Adipose-derived stem cells (ADSCs) form a sheet structure under the effect of L-ascorbic acid.

**Figure 3 bioengineering-10-00574-f003:**
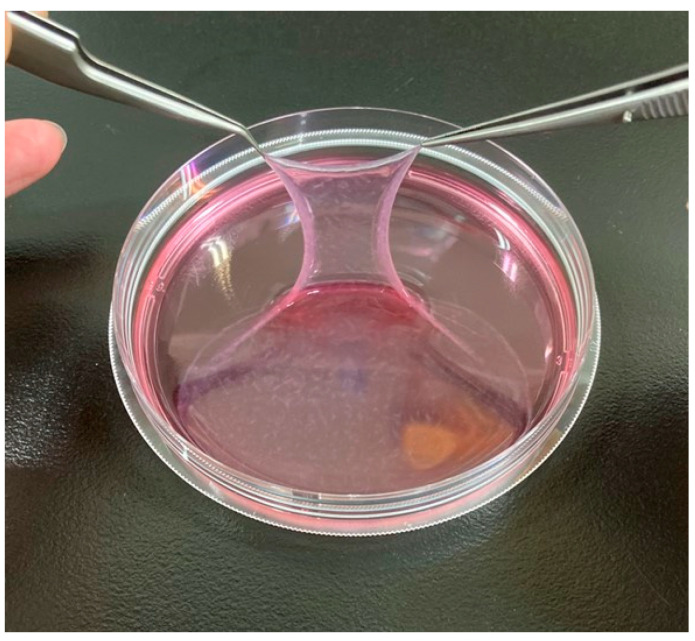
PRP-ADSC sheet. Chondrogenic differentiation of ADSCs was induced using PRP and formed a sheet structure using ascorbic acid. The resulting sheet structure was more robust than the ADSC sheet.

**Figure 4 bioengineering-10-00574-f004:**
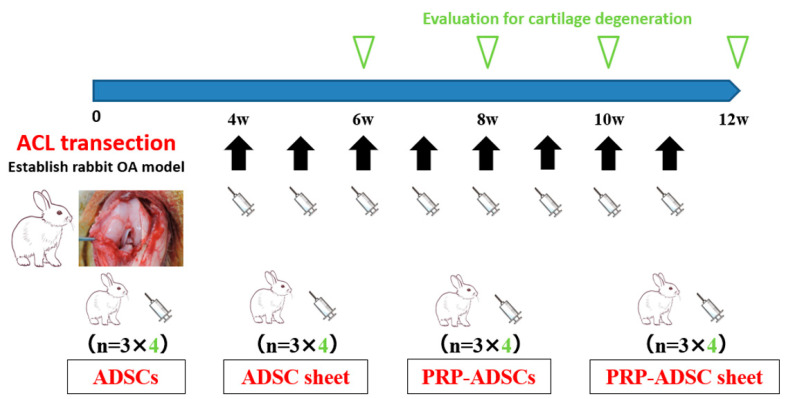
Schematic representation of the study design. Cells or cell sheets from each group were intra-articularly injected weekly from four weeks after ACL transection; for the four groups, three rabbits were sacrificed at 6-, 8-, 10-, and 12 weeks and six knees were collected as specimens at each time point.

**Figure 5 bioengineering-10-00574-f005:**
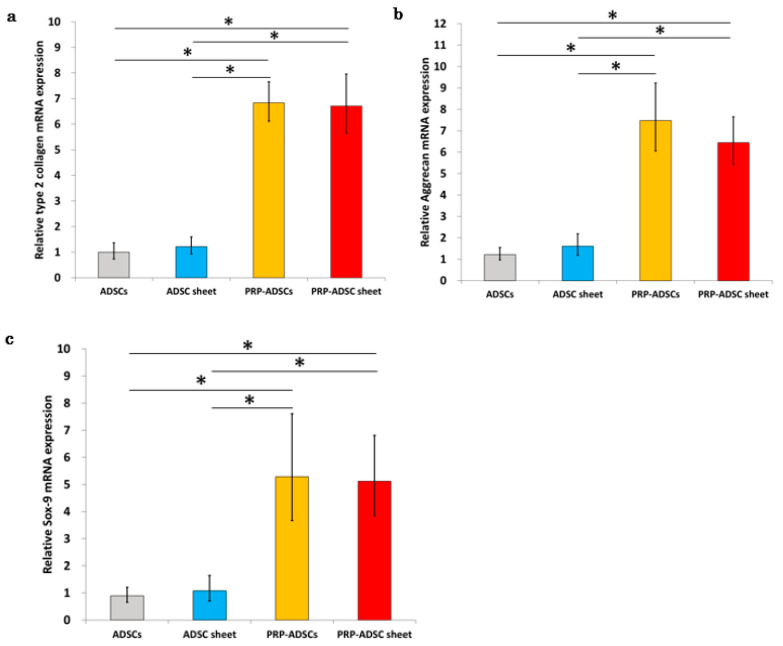
Real-time quantitative reverse transcription–polymerase chain reaction (RT-PCR). Relative expression of type 2 collagen (**a**), aggrecan (**b**), and Sox-9 mRNA (**c**) in the four groups. All values were normalized to the level of the GAPDH gene. All statistical analyses were performed using Welch ANOVA followed by Games-Howell post-hoc tests (*n* = 10 specimens per group). The error bars are defined as the standard error of the mean. * *p* < 0.05.

**Figure 6 bioengineering-10-00574-f006:**
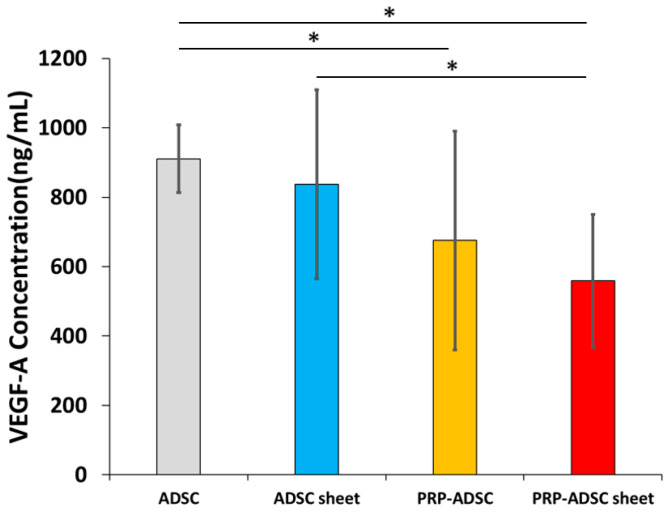
The concentration of vascular endothelial growth factor-A in the culture supernatant. All statistical analyses were performed using Welch ANOVA followed by Games–Howell post-hoc test (*n* = 10 specimen per group). The error bars are defined as the standard deviation of the mean. * *p* < 0.05.

**Figure 7 bioengineering-10-00574-f007:**
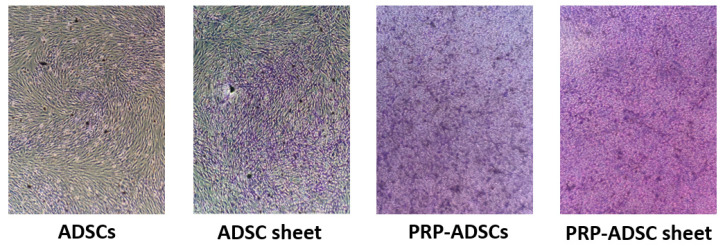
Toluidine blue staining. Stronger purple staining is observed in PRP-ADSCs and PRP-ADSC sheet groups than in the other two groups.

**Figure 8 bioengineering-10-00574-f008:**
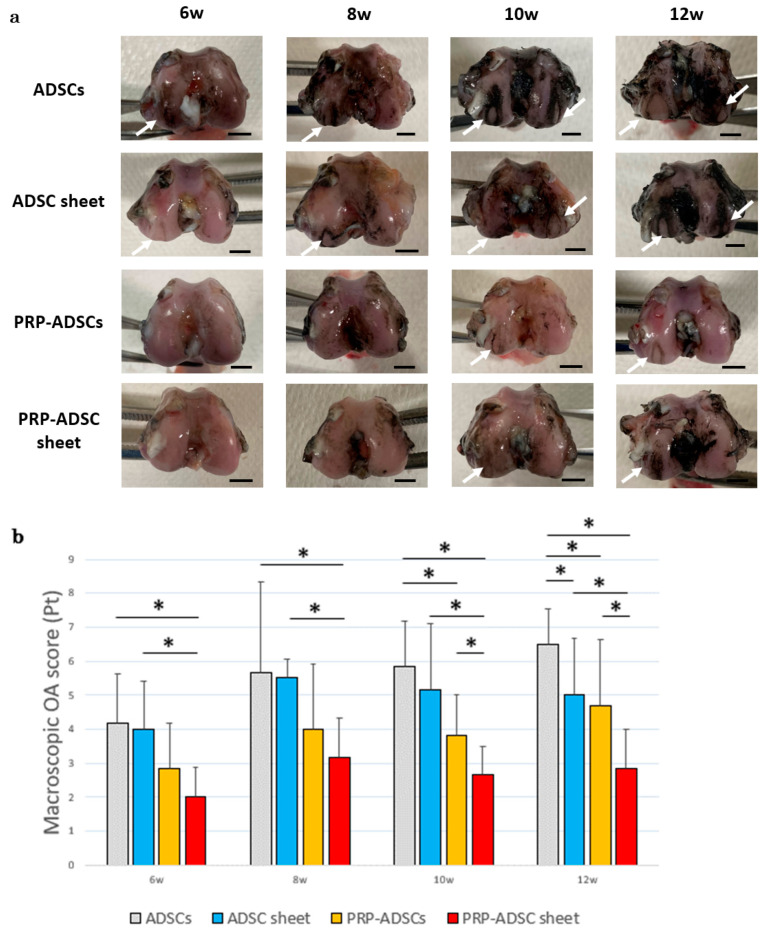
Macroscopic OA evaluation. (**a**) Representative findings of femoral condyles stained with Indian ink at each time period. White arrows indicate cartilage erosion. Scale bar = 5 mm (**b**) Macroscopic OA scores of ADSC sheets and control groups (*n*  =  6 specimens per group). All statistical analyses were performed using Welch ANOVA followed by the Games–Howell post-hoc test. The error bars are defined as the standard deviation of the mean. * *p* < 0.05.

**Figure 9 bioengineering-10-00574-f009:**
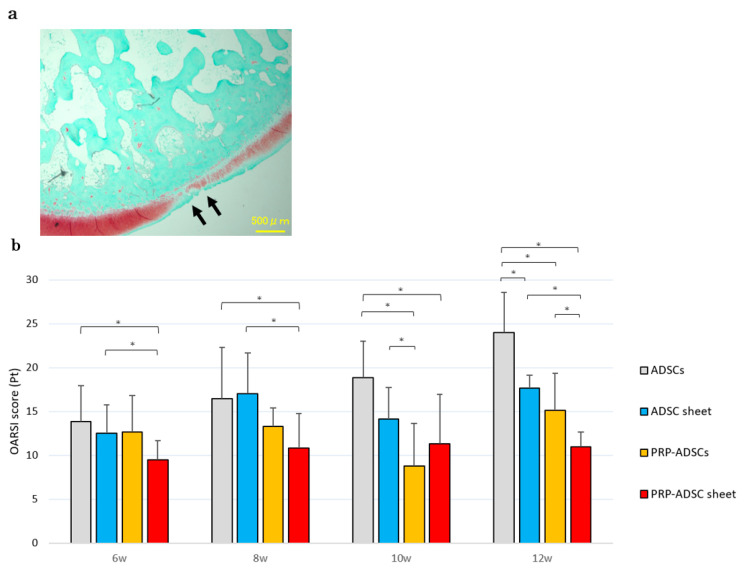
Histological OA evaluation. (**a**) Representative safranin-O-stained tissue sections. Cartilage erosion is indicated by black arrows. Scale bar = 500 μm. (**b**) OARSI OA scores. Six specimens were analyzed per group. All statistical analyses were performed using Welch ANOVA followed by the Games–Howell post-hoc test. The error bars are defined as the standard deviation of the mean. * *p* < 0.05.

**Figure 10 bioengineering-10-00574-f010:**
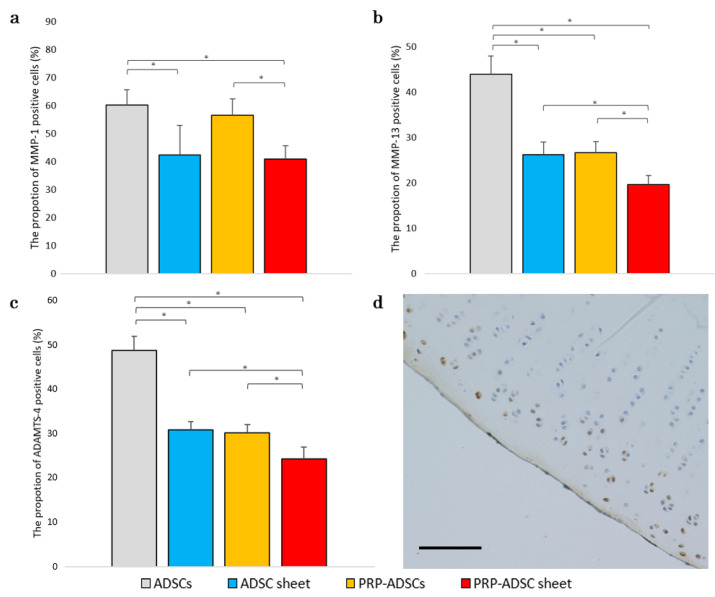
Immunohistochemical analysis for MMP-1 (**a**), MMP-13 (**b**), and ADAMTS-4 (**c**) was performed on specimens 8 weeks after ACLT. The proportion of positive cells is shown (*n*  =  6 specimens per group). The chi-square test was used to evaluate the comparisons of proportions among the groups. The error bars are defined as the standard deviation of the mean. * *p* < 0.05. (**d**) Representative specimens. Brown-stained cells indicate positive cells. Scale bar = 500 μm.

**Figure 11 bioengineering-10-00574-f011:**
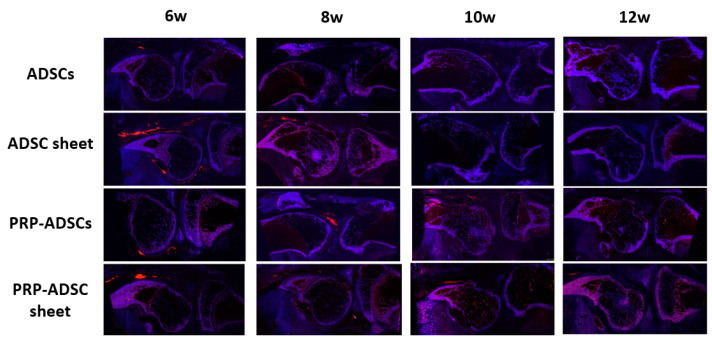
Qualitative evaluation using DiI-labeling was performed to observe the fate of injected ADSCs. Red-stained areas indicate DiI-positive areas. Although the DiI-labeling staining gradually faded over time due to cell loss, cell division, and pigment migration to the surrounding tissues, the PRP-ADSC sheet group showed the longest residual fluorescent staining.

**Table 1 bioengineering-10-00574-t001:** Macroscopic OA score. Both the medial and lateral femoral condyles were individually scored. The scores for both condyles were combined to obtain a cumulative score for macroscopic OA. The maximum score possible is 10 points.

Grade	Gross Finding	Score
1	intact articular surface	0
2	minimal fibrillation	1
3	overt fibrillation	2
4a	erosion of 0 to 2 mm	3
4b	erosion of 2 to 5 mm	4
4c	erosion of >5 mm	5

## Data Availability

The datasets used and/or analyzed during the current study are available from the corresponding author on reasonable request.

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
