# Peer review of "Chondroprotective Effects of Chondrogenic Differentiated Adipose-Derived Mesenchymal Stem Cells Sheet on Degenerated Articular Cartilage in an Experimental Rabbit Model"

_bioengineering, 2023, doi:10.3390/bioengineering10050574_

Round 1

Reviewer 1 Report

The manuscript presents novel data on adipose-derived stem cell use for rabbit cartilage regeneration during OA. ADSCs were used in four different groups, by applying ADSCs alone, stimulating them with ascorbic acid in cell sheets and inducing their chondrogenic differentiation with PRP in/without cell sheets.

Even though the manuscript presents significant amount of valuable data, I suggest major revisions to improve text and data quality.

My major concerns are:

·      Why only ascorbic acid was used to induce chondrogenesis in ADSCs? Why not complete chondrogenic medium with ITS, TGF, proline, etc. as a control?

·      The quality of the figures and graphs should be improved

Line 11: Adipose derived stem cells are not new in OA research, but have been studied in pre-clinical and clinical studies for many years.

Line 15: abbreviation to osteoarthritis should be used as mentioned earlier.

Line 39: Reference missing.

Line 44: Clarify the meaning of the sentence.

Line 53-54: Rephrase and clarify the sentence: bone marrow stem cells and etc are not the sources for regenerative stem cells, but are cells themselves. 

Line 64: Reference missing.

Line 68: typo – sheeted?

Line 120: abbreviation for Dulbecco’s modified Eagle’s medium required, as it was presented in line 117.

Line 123: CO2 typo, 2 should be subscript.

Line 125: What is collection density? This sentence needs clarification.

Line 130: 1 x 106 cells/dish – 6 should be superscript.

Line 142: “An” is a non-appropriate article to start this sentence.

Line 224-227: this sentence should be rephrased, the intensity of purple colour doesn’t indicate the presence of chondrocytes, but presence of negatively charged proteoglycans, which are being stained.

Line 229: I didn’t understand the method well – why were cell sheets dissolved before implantation? Experimental protocol should be clarified – were there any control rabbits with no ACL surgery, but sham?

Line 339: Schema is not an appropriate word here.

Line 248: osteoarthritis requires abbreviation.

Line 284: The observers were blinded? This sentence needs to be revised.

Line 314: Figure 5 presents relative gene expression as compared to house keeping genes? The legend should include this information.

Line 331-332: The fact that ADSC-PRP secrete lower amounts of VEGF as compared to ADCS only doesn’t make them potential to repair articular cartilage. This sentence is “too loud” and needs to be rephrased.

Line 334: Legend needs to be improved, it is not clear that VEGF concentration was measured in cell supernatants.

Line 349: Is there any positive control for Figure 7? Like cartilage sample stained with Toludine blue? It is very tricky to present results based on Figure color intensity, as the quality of the photos is not very good and it is hard to see cells and extracellular matrix around them. I suggest to improve the quality of the photos, maybe even add bigger magnification of the samples. The staining quantification and presenting in graphs would also be very helpful to compare GAG quantities among the samples.

Line 372: Figure 9 – is this the representative photo of histological section? Why others are not displayed?

Line 405: Stem cell therapy is not new for OA.

Line 428: Type II collagen, Aggrecan – should be lowercased.

English language has to be revised and improved. Some sentences are hard to understand.

Author Response

Thank you for your comments.

In this revision, the authors have tried our best to answer your comments, and we have added new information in the text, figure and table to clarify the points you raised.

We sincerely hope the reviewers will re-evaluate our revised manuscript.

Reviewer 2 Report

Figure 5.  SDs between groups, especially in panels b and c appear unequal.  Tukey’s test requires equal SD, but Games-Howell (GH) post hoc test does not.   Suggest GH be used for all post hoc multiple comparisons.

Methods section says error bars are SD, but figure 5 says error bars are SEM. Either correct figure bars or caption.

Figure 6.  SDs between groups appear unequal.  Tukey’s requires equal variance between groups, but Games-Howell test does not.

Line 342-3.  Sentence does not make sense.

 Figure 7.  The staining data is non-quantitative. This data is subjective. Suggest performing a quantitative test to measure glycans, such as an ELISA for aggrecan or via HPLC.  The authors are asked to add quantitative measures if they wish to include this figure in the paper.

Line 388-9.  Most significant downregulation…. This does not make sense.  Things are significant or not.  There is no more or most significant.  You set significance at p < 0.05, here.

Figure 8 and 9.   Are means and SDs shown (as specified in methods)? What statistical tests were used?

Figure 10.  Chi-square test should be used to compare proportions not ANOVA.

 Line 398.  more viable cells. Subjective report of information, not quantitative.  Please add quantitative measures if you wish to include this figure in the paper.  Aside, DiI is a membrane dye.  It stains intact membranes.  If transplanted cells undergo many cell divisions over weeks, the dye staining will “disappear”, too, due to dilution. Please revise descriptive language.

Minor: 

Line 403.  Two periods (typo)

The English is fine.  A few awkward phrases were found, but the text was understandable.   English is not an issue here. 

Author Response

(The authors gave the same response as above.)

Reviewer 3 Report

This study has potential for development of a more effective method of using adipose stem cells (ADSC) for the treatment of osteoarthritis. It employs an appropriate rabbit OA model and evaluates the potential of increasing chondrocyte differentiation by intra-articular injection of ADSC in a sheet structure formed with ascorbic acid in addition to treatment with PRP. This study adds to the research in the field with its use of the ADSC sheets and PRP and with its evaluation of levels of VEGF expression in these cultures.  The data appear to be significantly valid until it is noted, as stated by the authors in their own admission of limitations that the number of cells injected into the animal site of injury is not standardized for the same number of cells under the various conditions of cell culture. The only time there is any reference to the number of cells is at the initial time of seeding at the third passage.  Since PRP most likely, as has been extensively reported, produced cell proliferative and differentiation effects,  the final effects noted in the in vivo site can be largely because more cells were added which leads to more repair, not specifically because of any actual changes in the differentiation of the cells.  There is no doubt that additional studies are necessary with the same number of cells injected before the true nature of the effects observed can be evaluated. 

Author Response

(The authors gave the same response as above.)

Reviewer 4 Report

Generally a well written paper with good research design and controls.

Main criticisms:

In section 2.9.1 were the cell sheets disrupted with collagenase or disease prior to injection, or was an area of sheet injected? It should stated how the cells were counted and aliquoted for injection, as those in the sheets would be difficult to count and inject with just PBS washing;

Figure 11 is hard to assess. It needs further detail in either the legend or text as to the coloured areas visible. Also, the images seem to be from one 'group' only. Section 2.9.5 needs more detail as to the microscopy used. Also, it is difficult to determine whether the loss of DiI fluorescence is due to cell loss, cell division or dye movement to surrounding tissues. In the discussion, the lack of distinction between the OA knee and the healthy knee needs further emphasis.

Author Response

(The authors gave the same response as above.)

Round 2

Reviewer 1 Report

I thank the authors for their efforts correcting the text. The manuscript has been sufficiently corrected and can be accepted for publication.

Reviewer 3 Report

Revision is acceptable.